# A Scalable Solution for Node Mobility Problems in NDN-Based Massive LEO Constellations

**DOI:** 10.3390/s26010309

**Published:** 2026-01-03

**Authors:** Miguel Rodríguez Pérez, Sergio Herrería Alonso, José Carlos López Ardao, Andrés Suárez González

**Affiliations:** atlanTTic Research Center, Universidade de Vigo, Maxwell s/n, 36310 Vigo, Spain; miguel@det.uvigo.gal (M.R.P.); jardao@det.uvigo.es (J.C.L.A.); asuarez@det.uvigo.es (A.S.G.)

**Keywords:** NDN, LEO, mobility

## Abstract

In recent years, there has been increasing investment in the deployment of massive commercial Low Earth Orbit (LEO) constellations to provide global Internet connectivity. These constellations, now equipped with inter-satellite links, can serve as low-latency Internet backbones, requiring LEO satellites to act not only as access nodes for ground stations, but also as in-orbit core routers. Due to their high velocity and the resulting frequent handovers of ground gateways, LEO networks highly stress mobility procedures at both the sender and receiver endpoints. On the other hand, a growing trend in networking is the use of technologies based on the Information Centric Networking (ICN) paradigm for servicing IoT networks and sensor networks in general, as its addressing, storage, and security mechanisms are usually a good match for IoT needs. Furthermore, ICN networks possess additional characteristics that are beneficial for the massive LEO scenario. For instance, the mobility of the receiver is helped by the inherent data-forwarding procedures in their architectures. However, the mobility of the senders remains an open problem. This paper proposes a comprehensive solution to the mobility problem for massive LEO constellations using the Named-Data Networking (NDN) architecture, as it is probably the most mature ICN proposal. Our solution includes a scalable method to relate content to ground gateways and a way to address traffic to the gateway that does not require cooperation from the network routing algorithm. Moreover, our solution works without requiring modifications to the actual NDN protocol itself, so it is easy to test and deploy. Our results indicate that, for long enough handover lengths, traffic losses are negligible even for ground stations with just one satellite in sight.

## 1. Introduction

During the last few years, the number of massive Low Earth Orbit (LEO) constellations providing terrestrial communication services has grown dramatically. These orbiting networks made of thousands of satellites can provide Internet connectivity to almost any place on Earth with sky visibility. Moreover, when coupled with Inter-Satellite Links (ISL), they can complement terrestrial networks, becoming a low-latency Internet backbone [1]. In these new interconnected constellations, satellites play two simultaneous roles. On the one hand, they are the access routers to the orbiting backbone while, on the other hand, they are also the core routers of the constellation transporting traffic among different geographic areas [2].

These orbiting routers travel at very high speeds relative to their grounded counterparts. The low altitudes of LEO orbits, just a few hundred kilometers, yield orbital periods in the order of a hundred minutes. To accommodate these, and to maintain connectivity, ground routers need frequent handovers between passing satellites every few minutes [3,4]. Certainly, achieving end-to-end connectivity between two ground peers when part of the routing path goes through the satellite backbone stresses common Internet routing and mobility protocols [5,6,7,8].

In traditional TCP/IP networks that provide virtual channels for networked applications, the network and the mobility procedures have to keep this channel stable in the face of the almost continuous topology changes [9,10]. In contrast, Information Centric Networking (ICN) networks do not rely on a virtual channel for communication. Instead, the network is able to address *data* directly and bring it to the requesting clients.

The sophisticated forwarding mechanisms of ICN networks, along with the motivation to explore alternative networking paradigms, have driven significant research into their applicability as the network layer for emerging LEO constellations. Among ICN proposals, Named-Data Networking (NDN) [11] stands out for its maturity. In particular, NDN usage in LEO satellite networks has been a hot topic of research lately [12,13,14,15].

The modus operandi of ICN networks, and of the NDN proposal in particular, inherently solves the problem of the mobility of the receiver (or *consumer* known as NDN). The sender (called the *producer*) is unaware of the consumer(’s) identity and location, and the data simply travels backwards through the same routers used by the data request. This works as long as the path back from the producer to the receiver remains unchanged from the initial request of the data to its final delivery. The case of producer mobility is more involved, as it requires a method for the network to locate its current location. NDN networks have a standardized mechanism for this problem, the Kite protocol [16], but it is not appropriate for the massive LEO scenario.

Unfortunately, mobility in massive LEO constellations, with their frequent handovers involving a great number of mobile nodes (every ground node is mobile from the point of view of the satellite network), is not completely solved by the aforementioned approaches. Producer mobility in satellite networks presents unique challenges. Common mobility solutions, like Kite, depend on *immobile anchors* (akin to home agents in IP mobility) to maintain routes between consumers and mobile producers. However, in massive LEO constellations, placing these anchors on the ground is impractical, as ground nodes themselves experience frequent handovers and cannot provide the required stability. Alternatively, assigning satellites as immobile anchors introduces new issues: it is unclear which satellite(s) should serve as anchors for each ground node, and the dynamic nature of satellite orbits means that the network topology between anchors and ground nodes changes constantly. As a result, routing through such anchors often leads to suboptimal, longer paths and degraded performance, undermining the intended benefits of the approach. In the case of *consumer mobility*, the return path of data packets to the *consumer* is disrupted each time the consumer changes its access satellite, preventing the data from reaching the receiver.

In this article, we propose a comprehensive solution for the mobility problem of NDN-based LEO satellite networks. Our solution is able to maintain connectivity between any pair of ground stations during handovers in a scalable manner and without requiring changes to the NDN specification [17]. We will obviate the routing of the requests from the consumer to the first ground station and from the second ground station to the producer, as those are already solved by the usual routing procedures. We also provide a scalable solution for locating the most appropriate ground node for a given producer.

Our solution solves the producer mobility problem in a scalable manner and without relaying on *home agents* or *immobile anchors* like the existing solutions in the literature. In fact, producers do not need to carry out any extra management procedure to register their new network location after a successful handover. This is due to the fact that the solution does not try to be generic, but is adapted to the specific scenario of massive LEO constellations. Additionally, all the complexity related to the producer movement is hidden from the satellite nodes, as they are assumed to be more resource-constrained than the ground nodes. In the same spirit, the only ground nodes involved in the mobility procedures are the ground stations. While this is a departure from the end-to-end principle, it simplifies the solution and eases deployment. Finally, the proposed solution can be implemented with the facilities present in the current NDN version. As far as we know, our solution to the producer mobility problem is completely novel. For the consumer mobility part of the solution, we were able to build upon some of the ideas already introduced in [18].

The rest of this paper is organized as follows. Section 2 provides the background and a short introduction to NDN fundamentals. Section 3 describes the scenario and its associated mobility challenges. Then, in Section 4, we describe how to solve the producer mobility problem. The consumer mobility problem is dealt with in Section 5. Section 6 contains the experimental results. Finally, Section 7 provides a discussion on the proposed techniques and results, to lay the conclusions in Section 8.

## 2. Basic NDN Concepts

In traditional IP networks, when an IP node sends data to a peer, it simply adds the appropriate network header with the corresponding destination IP address to the data, and the network delivers it to the desired peer(s). NDN works differently. In NDN networks, communication follows a pull-based strategy in which *consumer* applications request data from the network that is ultimately provided by a *producer* application, usually residing on another host. This results in two different packets. The first one, demanding a *named* piece of content, is the *Interest* packet that arrives to an NDN node through one of its interfaces (*faces* in NDN parlance). The second one, containing the response, is the *Data* packet.

The Interest packet contains, at least, the *name* of the requested content. NDN nodes can implement a cache of popular content inside their Content Store (CS) (see Figure 1). Then, any NDN node that has a copy of the requested data (as identified by its name) can return it directly into a Data packet. To facilitate forwarding, names in NDN follow a hierarchical structure, so that routers can direct the Interest packet towards a producer based on a prefix of the name. When an NDN router decides to forward an Interest packet, it annotates its name and incoming interface in the Pending Interests Table (PIT).

This serves two purposes: firstly, it will be used to forward the Data packet back to the requesting consumer; secondly, it aggregates identical requests from different incoming interfaces, enabling seamless multicast data transmission. To determine the outgoing face, the node uses the Forwarding Information Base (FIB), which is a routing table mapping names to outgoing interfaces. However, before indexing the table using the requested named content, NDN nodes use information that can be provided in the Interest packet itself. Interest packets may carry a *Forwarding Hint*, which is a list of names associated with some topology information that are easier to route than the actual name of the content.

The response with the requested content travels in a *Data* packet. When a node receives Interest and has (or can generate) the named data, it puts it into a Data packet and forwards it to the interfaces listed in the PIT for that name. Intermediate nodes that receive the Data packet can store its contents in their CS to provide a cache for later requests for the same name. Eventually, the Data packet reaches all demanding consumers using the information stored in the PIT of the intermediate routers.

## 3. Problem Description

Massive LEO constellations are composed of thousands of satellites organized in shells of multiple orbital planes, each one with an equal number of equi-spaced vessels. In recent deployments, each satellite can establish four stable ISL: two with the preceding and succeeding satellites in their common orbital plane (V-ISLs) and two with the closest satellites in the two adjacent neighboring planes in the same orbital direction (H-ISLs). Note that these constellations mostly employ circular orbits in a Walker-delta or Walter-start topology [19] at a given altitude. As the velocity of each satellite is determined by its altitude, so it is its visibility time from a given latitude. Thus, the minimum handover frequency decreases with the altitude. Similarly, the number of circular orbits and the quantity of satellites in each orbit affect the density of the constellation and, correspondingly, the number of satellites that can simultaneously serve a given ground station. In these designs, the satellites in adjacent orbital planes are arranged to maintain consistent neighboring relationships. This means that a satellite in one plane will have a corresponding satellite in the adjacent plane that it is *paired* with [20]. Even in more sophisticated constellations, like Starlink, that do not follow a strict Walker-delta design, satellites can form four stable ISL with their closest neighbors [21]. Certainly, H-ISL links may not be available in the polar regions if the constellation has a very high inclination [22]. However, we will disregard this eventuality to keep the scenario simple, as it does not affect our mobility solution.

Figure 2 shows an example of such a constellation.

To avoid cluttering the figure, it represents a very small constellation with just four different orbital planes, each one with a distinct color and, with dotted lines, the ISL links of satellites in the *blue* orbital plane. Due to its small size, the H-ISL links shown cover long distances, but most commercial designs for massive LEO networks are a couple of orders of magnitude denser. Also note that the satellite in a neighboring plane may not coincide with the closest one at a given time, since the latter may be traveling in the opposite direction. For instance, the blue satellites in the figure establish links with the green and red satellites, but not with the yellow ones.

The topology formed by the ISL links does not change as the satellites orbit their planes, because the relative locations of the satellites in their planes and the positions of the planes remain quite stable. Thus, if we ignore the physical location of the satellites while they travel across the globe, and focus just on their interconnections, the actual network topology corresponds to a grid, like the one represented in Figure 3.

When all the links are operational, forwarding traffic between any two satellites becomes straightforward, provided their grid locations are known: simply forwarding packets to a neighboring satellite closer to the destination guarantees the minimum number of transmissions. However, if latency is a concern, or when not every link is available, there are more elaborate proposals that consider the propagation delay [23] or, in NDN networks, the location of the caching nodes [24,25]. In any case, the inherent topological structure of the satellite backbone stays relatively stable.

The main challenge arises from the interaction between ground nodes and their orbiting counterparts. There are frequent handovers between ground stations and satellites, as the latter remain in sight of a given ground location for just a few minutes at a time. Moreover, it is unfeasible to determine which satellite(s) is(are) being tracked by a ground station without additional information, as in a massive LEO constellation, the ground station may have tens of candidate satellites to choose from.

We illustrate the whole transmission process with the help of the scenario depicted in Figure 4.

It shows a communication path between a consumer and a producer traversing the satellite backbone (we assume that all consumer and producers are Earth-bound). Before reaching the producer, the Interest packet sent by the consumer must first arrive to a ground station acting as a gateway node between the ground and the satellite network section, the Consumer Gateway (C-Gw). Then, it traverses across the satellite network until it reaches an appropriate ground station closer to the producer, i.e., an exit gateway that we are calling the Producer Gateway (P-Gw). Afterward, it proceeds normally towards the producer. Note that, due to the global coverage of the satellite network, there should not be more than one satellite segment in the path between the consumer and the producer. The response Data packet follows the reverse path. We are not concerned with how the gateways are reached by the consumer and the producer, as there is no added complexity due to the satellite section. Instead, we will focus on how the C-Gw and the P-Gw can communicate and how to identify the P-Gw for a given Interest.

We assume that at any given time, both gateways can communicate with one or more satellites, but due to the high speed of the satellites, their respective access satellites change every few minutes.

We will first tackle how the C-Gw communicates with the P-Gw (producer mobility).

## 4. Producer Mobility

The job of the C-Gw is to forward the Interest packets it receives from other devices to the appropriate P-Gw so that they eventually reach the corresponding producers. This entails several steps:The C-Gw determines the identity of the P-Gw for each Interest;Then, it addresses the Interests to the satellite currently being used by the P-Gw;Finally, the satellite network forwards the Interests to the appropriate satellite efficiently.

Notice that the satellites themselves only have to deal with forwarding between satellites, but are freed from tasks like locating the appropriate exit point. This should remove the computational load from them and place it on the ground gateways, which have more computing and power resources.

### 4.1. Identifying the P-Gw

Identifying the appropriate P-Gw means obtaining the identity of the ground station closest to the actual producer. We name each P-Gw as <sat_prefix>/<P-Gw>, with <sat_prefix> being the identifier assigned to the satellite constellation operator, and <P-Gw> being a different identifier for each gateway. Each ground station will be able to reach a certain amount of ground prefixes. This information, in the form of a set of correspondences between a P-Gw identity and a prefix, must be made available to the rest of the ground stations so that C-Gw can target their Interest packets accordingly.

This information, which is not subjected to frequent variations, can be propagated with the help of a distributed database application, similar to the DNS, and managed by the satellite network operator. Essentially, gateways will use this application to announce their own names along with the set of prefixes that they can reach. When a consumer needs to forward an Interest for a prefix, it can retrieve the name of the remote gateway from the service. For example, NDNS [26] and State-Vector Sync (SVS-PS) [27] are well-known methods for distributing information on NDN networks, and both mechanisms can be easily adapted to this scenario.

### 4.2. Addressing Interests to the P-Gw Access Satellites

Once the identity of the proper P-Gw has been determined, the next step is to forward the Interest packet to a satellite that is currently being tracked by the P-Gw. Then, when the Interest packet finally arrives at this tracking satellite, it delivers the packet to the P-Gw that forwards it according to the normal procedure along the earth-bound segment of the route.

The first part of the problem is finding an actual satellite the P-Gw is tracking at the time of the Interest transmission. One possible solution would be to use (or reuse) a distributed service mapping satellites to ground stations. However, whereas the mapping between P-Gw and reachable prefixes is relatively stable, the relationship between ground stations and access satellites is continuously in flux. Reusing such a system for this objective would entail constant updates of the service database, causing excessive traffic and computational load. Instead, we propose encoding the geographic location of the P-Gw, as the <P-Gw> suffix of its name. The geographic location of ground gateways is stable, and can be used to identify the set of satellites that they have at sight at any given time. This is a straightforward task, assuming that the C-Gw has access to the constellation data. This information can be provided in real time by the satellite network operator. For instance, it can be provided to the ground station by the access satellites themselves. It also should not need high accuracy, just enough to have a rough estimate of the satellites possible serving a given region.

The possibilities for encoding the node location as its suffix are plentiful, but we have decided to rely on Open Location Code (OLC) [28] as it provides a natural way to obtain arbitrary precision, allowing us to address gateways in an area or pinpoint their locations with as much accuracy as needed. In essence, OLC can encode any region of the planet as an alphanumeric string of variable length. The longer the string length, the smaller the area covered by the region. For instance, OLC would encode the location of a P-Gw located in our campus (42.169938N, W8.687812) as *plus code* 8CJH5896+XV, with a precision of about ±3 m. So a C-Gw that just wants to contact any gateway near the region would address interests to <sat_prefix>/8C/JH, or near the campus to <sat_prefix>/8C/JH/58, etc. Then, a gateway called <sat_prefix>/8C/JH/58/96/XV, inside our building, would only answer if the requested name contains its own name as a prefix. Please, note that the area covered for a given prefix length decreases with the latitude, as the longitudinal lines grow closer. C-Gw should take this into account when determining the needed accuracy in locating a P-Gw.

The second part of the problem is getting the Interest packet forwarded to the set of satellites instead of directly to the P-Gw. Thankfully, NDN already provides a mechanism for this. Normally, in NDN networks, an Interest is forwarded according to the name of the requested content. However, there exists the possibility of providing additional information to NDN routers as *Forwarding Hints* included in the Interest packet. These *Forwarding Hints* consist of a set of *delegated* names to help the forwarding procedure. In fact, the predefined routing *strategies* use the first reachable *delegation* instead of the named content to forward the Interest packet. In NDN networks the routing procedure, called *strategy*, can be modified at any router and even vary according to the prefix. In our solution, the satellites use a modified routing strategy for the sat_prefix that employs all the *delegations* listed in the *Forwarding Hint* to forward the Interest.

So, once the set of satellites that it has at sight at any given time has been determined, the C-Gw adds their names as a list of *delegations* in a *Forwarding Hint*. Additionally, the name of the P-Gw is also included in the *Forwarding Hint*. This allows the actual serving satellite to identify the ground station to which it must forward each Interest packet.

### 4.3. The Forwarding Procedure

When an NDN satellite receives an Interest that cannot be satisfied with its CS contents, it selects the follow-up forwarding actions according to the strategy registered for the named prefix. The forwarding actions must ensure that the Interest packet reaches all the satellites identified by the C-Gw as possibly being in use by the P-Gw. However, it is also important that this forwarding process minimizes the number of copies of the Interest packet to keep the network load as low as possible.

We will assign a name to each satellite to facilitate the forwarding procedure. Since the satellite shell topology is essentially a two-dimensional grid, each satellite can be easily identified by its own coordinates in the grid. Thus, each satellite in the network will be given a name of the form <sat_prefix>/<plane>/<index>. The forwarding procedure consists of selecting which subset of the delegation list provided by the Forwarding Hint is going to be served by each ISL link. Then, the Forwarding Hint of each Interest copy forwarded to a neighboring satellite is updated to carry just the proper subset of delegations. (Please note that this procedure even applies to signed Interest transmissions. In NDN, the Interest signature does not include the Forwarding Hint, so forwarding nodes are free to alter it if necessary without compromising the security of the communication.) The grid-like topology of the satellite constellation used to illustrate the mobility problem in this paper facilitates the straightforward selection of valid ISL links. Certainly, in the case of more complex topologies or link failures, a more elaborate routing algorithm should be used instead to select them.

We will use the example shown in Figure 5 to illustrate the procedure. The Interest arriving at node (10,10) carries a delegation list consisting of three different names, corresponding to nodes (12,3), (14,15) and (10,17). The first step is deciding which neighbors are closer to each destination than the current node. In this grid-like topology, this corresponds to the 1-norm in modular arithmetic. Take delegation (12,3). Its distance from node (10,10) is (12−10)+(10−3)=9, whereas from neighbor (11,10) the distance is (12−11)+(10−3)=8, so (11,10) is closer to the destination, hence it is a valid next-hop. Following the same reasoning, delegation (10,17) shall use (10,11) as the next hop. So we need two independent transmissions for these two delegations. Finally, delegation (14,15) has two equally valid next hops: (11,10) and (10,11). To minimize the number of transmissions, the delegations that can be included in more than one transmission are placed in the Interest packet with the highest number of delegations. In this example, both carry just one, so the selection is random. In the figure, it was included in the one towards (11,10).

The exact procedure is summarized in Algorithm 1. Its input is the set of all the delegations present in the Forwarding Hint (D). The first line initializes four subsets ($D_{F},D_{A},D_{P}\;\mathrm{and}\;D_{S}$) for holding the delegations to be served by each ISL link (in the Fore, Aft, Port and Starboard directions). Lines 3–6 implement a greedy algorithm for selecting the appropriate forwarding set. In line 5, *s* represents the current satellite; L(·) is the location of the given satellite; $N_{h}\left(s\right)$ is the node across link *h* from *s*; and, finally, ‖·‖ is the 1-norm, which is used to calculate the distance between two nodes in the grid-like network topology. So, at line 5, the procedure just adds delegations to a set if the neighbor across the link is closer to the given delegation than the current node. The second part of the procedure just removes redundant transmissions. Lines 8–10 empty delegation sets if all their delegations can also be carried out by the rest of the delegation sets. Lastly, if there are any remaining delegations served by more than one delegation set, they are removed from all but the largest set (lines 11–15).
**Algorithm 1** Procedure for Interest packet dissemination.
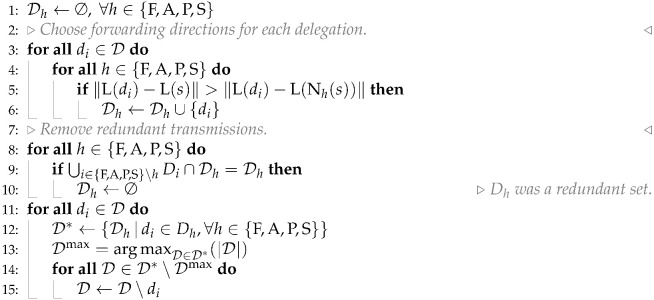


Eventually, when the Forwarding Hint includes the current satellite, and so as it is a candidate to be serving the P-Gw, it will forward the Interest packet through the corresponding satellite-to-ground interface to reach the P-Gw.

It is possible to adapt this procedure to route around a few possible link failures or even to consider complete seams in the satellite topology. In the general case, reaching the targeted satellites can be achieved with a regular routing algorithm that is able to provide the appropriate next hop along the path.

### 4.4. Performance Optimizations

When the C-Gw wants to send more than a few packets to the producer, to reduce unnecessary transmissions, the C-Gw should only include one actual access satellite of the P-Gw in the delegation list of the forwarded Interest packets and not every potential satellite at its sight. This can be accomplished by previously asking the P-Gw for the name of a current access satellite. As the name of the P-Gw is also known (<sat_prefix>/<P-Gw>), this can be retrieved by sending an Interest packet directly addressed to it with an appropriate name, e.g., <sat_prefix>/<P-Gw>/access. The P-Gw shall respond with a *Link* object containing the name of one of its current access satellites. A Link object is just a specialized Data packet containing a collection of names and preferences to be used directly as a Forwarding Hint (the delegation set). Additionally, a freshness period for this Link object is set so that it expires before the P-Gw considers a new satellite to be employed.

Since this Link object is a Data packet, it is stored in the CS of the C-Gw (and in that of any other NDN node in the path from P-Gw to C-Gw that decides to do so). So, if during its freshness period, the C-Gw sends additional Interest packet requesting this Link object, they will be satisfied from the local CS without requiring a new network transmission. When the freshness period expires, any new Interest packet for the Link object will eventually reach the P-Gw to receive updated information. This provides an automatic mechanism for updating the Link object.

However, note that if the C-Gw waits until the Link expires to send a new Interest to query the access satellite, it would need to address this new Interest to every possible satellite at sight by the P-Gw again. To avoid this, we propose the following strategy that should be valid even in the case of P-Gw with single-beam antennas limited to tracking only a single satellite at a time:The P-Gw generates the Link object with a freshness period smaller than the expected remaining time with the selected access satellite at sight. When the freshness period expires, the P-Gw starts a handover process with a predefined duration *H*.When the Link expires in its CS, the C-Gw sends a new Interest requesting a new Link, using as forwarding hint the just-expired delegation list (i.e., one with a single entry corresponding to the still current access satellite). This should arrive at the P-Gw before the handover period (*H*) finishes.When the P-Gw receives the Interest packet during the handover period, it generates a Link with two names, those of the current and next access satellites. The freshness period should be small, but longer than the time remaining until the handover period ends. This time can be equal, for example, to the handover length *H*. In the case of multi-beam antennas, the link can contain just the name of any other satellite that is being tracked and the remaining visibility time as the freshness value.When the handover process is complete (after *H* seconds), the P-Gw switches to the new satellite and assembles the corresponding new Link object. Shortly thereafter, the temporary Link object in the C-Gw expires and the C-Gw requests the new Link object by sending an Interest addressed to the names contained in the just-expired one. This will arrive to both the old access satellite, which will not be able to forward it to the P-Gw, and to the new one. The new access satellite will succeed in forwarding it to the P-Gw that will answer with the proper Link object, valid for the next few minutes.

It must be noted that the selection of *H* does not have to be overly precise, only big enough to give enough time to complete the handover, but not too big to send too many unnecessary packets simultaneously to both targets. Being a time interval, it does not need any synchronization between the clocks of the participants. And, given its relatively short duration and lack of precision needed, it should be immune to any existing clock drift. Additionally, the optimum *H* value, i.e, the lowest one that avoids losses, is dependent on the delay between the C-Gw and the P-Gw, so, ultimately also on their locations and the satellite segment parameters. The C-Gw nodes could measure packet losses during handovers to tune the *H* value starting with a conservative guess.

Finally, in the case that the P-Gw is capable of tracking more than one satellite simultaneously, the procedure is greatly simplified. Before one of the satellites is about to stop being in sight, the delegation list is updated with the remaining satellites in sight and the freshness period is set according to the satellite with less remaining time in sight.

## 5. Consumer Mobility

When a C-Gw changes its access satellite, all pending Interests cannot be satisfied because the Data packets fulfilling them cannot reach the C-Gw after the change. Even though the geographical position of the C-Gw remains the same, from the point of view of the satellite network, its point of attachment has changed, and thus it has apparently *moved*.

As mentioned in the introduction, this issue has already been addressed by [18]. The solution presented in said work consists of retransmitting pending Interests to the previous access satellite. This is also achieved with the help of a Forwarding Hint. We can reuse the same approach but performing the retransmissions directly at the C-Gw, instead of at the consumer or at any other in-network node. This satisfactorily restricts the complexity of roaming operations to the nodes directly involved, i.e., the ground gateways and the satellites.

The procedure, depicted in Figure 6, is as follows.

Consider a C-Gw that has some pending Interests that were transmitted via its old access satellite (represented by steps 1 and 2 in the figure). The old access satellite had already filled an entry in its PIT for each pending Interest of the C-Gw through its ground interface. Then, it forwards each Interest to the rest of the satellite network through one of its ISL links. After a C-Gw handover:The Old Access Satellite (OAS) does not purge its PIT. If the pending Interests are fulfilled, the corresponding Data packets are stored in the CS.The C-Gw retransmits all its pending Interests to its Current Access Satellite (CAS), but with a Forwarding Hint that employs the OAS name as the only delegation. In the figure, these correspond to steps 3 to 5.The OAS updates its PIT adding the interface on which each Interest was received from the CAS to the list in the corresponding PIT entry. In the example, it is ISL3. Eventually, when it receives matching Data packets, it will forward them back to the CAS, according to the information in the PIT. Alternatively, for Data packets that have already arrived, it fulfills their Interests with the information from the CS.The CAS normally forwards the recovered Data packets to the C-Gw, which can then forward them to the consumers as usual according to the information stored in its PIT.

## 6. Results

To validate our approach, we have developed a simulation module for the NDNSim [29] that implements the described consumer and producer gateways [30]. In particular, the code implements (1) the required *strategy* used by the satellites to forward Interest packets according to their positions in the grid; (2) the consumer application used by the C-Gw to obtain the serving satellite(s) of the P-Gw; and (3) the producer application running on the P-Gw to respond to the Interests from the C-Gw. The application at the C-Gw is also responsible for performing retransmissions of pending Interests whenever there is a C-Gw handover, as detailed in Section 5. On top of this scenario, we run a stock consumer application that asks for named data packets at a constant rate. These Interests are satisfied by another stock producer application running on the P-Gw itself. We have not implemented, however, the service for querying the physical location of the P-Gw, since we assume that it is known beforehand.

The simulated satellite network has a design similar to that of the first group of the phase of Starlink [31], with 72 orbital planes and 22 satellites per plane, at an inclination of 53° and an altitude of 550 km. In each of the following experiments, the consumer and the producer exchange data for two and a half hours (10,000 s to be precise), so that there are many handovers on both the P-Gw and C-Gw sides. The P-Gw and the C-Gw have been situated at latitude 42° North, for no other reason that it corresponds to a densely populated area on Earth, and it happens to include the location of our lab. The distance between them is 5300 km. In order to test the effectiveness of the handover procedure, we consider a worst-case scenario in which ground gateways can only track a single satellite simultaneously. With these experiments, we want to observe how the mobility procedures affect data retrieval and the repercussions on the periods in which the communication is interrupted. In order to try the most demanding case for the mobility procedure, all ground stations are restricted to communicate with a single satellite at a time. We do not model the power constraints of the satellites as the additional load placed on them by our mobility solution (the transmission of some extra Interest packets at the beginning of communications and during handovers) is negligible.

### 6.1. Results at the P-Gw Site

We first analyze the results of the mobility procedures on the producer side. Recall that every time the P-Gw starts tracking a different satellite, the C-Gw has to obtain an updated delegation list that includes the identity of the new access satellite. To ease the transition, as detailed in Section 4.4, there is a handover period of configurable length *H* during which the C-Gw uses a delegation list consisting of both the old and new tracked satellites.

Figure 7 shows the percentage of P-Gw handovers in which at least one Interest packet was not answered for different values of the *H* parameter. Losses of Interest packets can happen for two different reasons that are distinguished in the figure.

The first situation happens when the handover length is too small. In this case, the C-Gw will not be able to obtain the updated Link object before the P-Gw finishes the handover procedure. So, some of the Interests sent by the C-Gw will reach the old access satellite, even if the P-Gw is not connected to it by their arrival time. This will also prevent the C-Gw from obtaining an updated delegation list. When the C-Gw sends the Interest querying for it, addressed to the previous P-Gw serving satellite, the handover will proceed, so the Interest will not ever reach the P-Gw. This cannot be avoided, not even by sending this Interest before. If the C-Gw were to send the Interest before the handover happened, by the time the Interest reaches the P-Gw, the handover process would not have been started yet, and so the Link object will consist of the satellite that is to be abandoned. When this happens, the C-Gw will keep on sending Interests to the old access satellite until they timeout. In any case, when the stranded Interest querying for the updated delegation list timeouts (1 s in our experiments to see it easily), the C-Gw will calculate the complete set of possible access satellites, obtaining a new delegation list, and the communication will resume, albeit in a less efficient manner, until the identity of the new access satellite can finally be determined, as per Section 4.4. In Figure 7, the traffic loss periods that end after a timeout are responsible for most of the traffic loss periods when the handover length is small. In fact, when the handover length is 0 s, all handovers suffer packet losses due to this cause. As the handover length increases, the percentage of handovers leading to timeouts diminishes. Indeed, timeouts occur only when the transmission delay between the C-Gw and the old access satellite is less than the handover length. On the right side of the graph, when the handover length is high enough, the C-Gw is always able to obtain the updated Link object and it never times out.

The second kind of loss depends on the relative distances, from the point of view of the C-Gw, to the old and new access satellites of the P-Gw. Note that Interest packets get lost when the new access satellite is closer to the C-Gw than the old one. As this is random, it happens in approximately 50% of the handovers, as shown in the figure. To see how this occurs, consider an Interest that is being directed at both the old and the new access satellite. When the new one is closer to the C-Gw, the Interest should reach it before it gets to the old one. However, for a few Interests, this happens *before* the handover has been completed, and so they are discarded by the yet-to-be accessed satellite. Then, if the actual handover is finalized just after this, and before these Interests reach the old access satellite, they are discarded by both. Note that, even though this situation can potentially happen in half of the mobility events, it affects only a few packets, since the duration of the event is very short.

To shed more light on the impact of P-Gw handovers on traffic loss, we also measured the average length of the traffic loss period in each handover for different values of the handover length parameter. The results are shown in Figure 8, both as the absolute time (left axis) and as the percentage of the time between two handovers (right axis). For small handover lengths, when most losses are due to timeouts, the loss length is close to the timeout value. In fact, without a handover period (i.e., H=0), it is slightly higher, because this length includes both the timeout and the time taken to get a new valid delegation list. However, as the handover length increases and fewer and fewer loss periods occur due to timeouts, the average length of the loss periods gets closer to 0. This corresponds with the loss events due to the relative positions of the old and new access satellites and, in these cases, the length of the loss periods is limited to the difference between their relative positions, which is usually very small. Notice also that, for any *H* value, the time period when losses may happen is always less than 1% of the time between consecutive handovers.

### 6.2. Results at the C-Gw Site

The consumer mobility procedure also affects traffic delivery. Whenever the C-Gw changes its access satellite, the return path of the Data packets serving pending Interests is broken. When the consumer mobility procedure detects the mobility event, it recovers the missing data re-sending the Interests through the new access satellite using a forwarding hint to route the Interests to the old access satellite. If the Data arrives at the old access satellite before the retransmitted Interest reaches it, it answers with a copy of the Data from its content store. If not, the incoming interface is added to the PIT entry to the requested content. When the Data finally arrives at the old access satellite, a copy is sent back to the new one and then arrives to the C-Gw.

To exemplify how this mechanism performs, we have simulated a scenario where a consumer sends one thousand Interests per second to a producer in the same LEO network topology as before. The purpose of the traffic rate is not to measure the network capacity, which is orthogonal to the mobility issue, but to capture the times when packets are either delayed, retransmitted, or lost due to mobility.

Figure 9 shows the arrival time of Data packets and the departure times of their corresponding Interest packets. At time 426.148 s, the access satellite is modified. Immediately, there is a gap in the arrival times, since the Data packets queued for transmission to the C-Gw at the old access satellite cannot be transmitted. Again, the C-Gw mobility procedure requests old pending data in the PIT of the C-Gw and, at time 426.155 s the re-requested packets arrive through the new access satellite. They arrive almost simultaneously because they had all already been received by the old access satellite, so, in this case, the round trip time between the C-Gw and the old access satellite via the new one is just 7 ms (an actual implementation may want to pace the retransmission of pending Interests to avoid traffic surges in the return path.) Note that all Data packets have been recovered, as there are no horizontal gaps between consecutive packets. Finally, at time 426.156 s, the first non-retransmitted request arrives, and the transmission goes on as normal.

## 7. Discussion

Any solution to the mobility problem of massive satellite communications must be both scalable and secure.

There are three distinct elements that participate in our solution: the consumer gateway, the producer gateway, and the satellites themselves. The strategy we have chosen to map NDN prefixes to P-Gw is akin to the one used on the Internet to map domain names to IP addresses, so scalability should not be a concern. Regarding the challenge of mapping the P-Gw to the set of possible satellites at its sight, identifying these satellites does not involve any communication, since it is carried out by the C-Gw using the ephemeris information of the constellation. However, the size of this set grows linearly with the constellation size. We do not believe that this is a serious issue, as only the first Interest packet between a C-Gw and a P-Gw addresses the whole set of possible satellites. Once the actual satellite has been determined, the rest of Interest packets address only one (sometimes two) satellites, regardless of the constellation size. Finally, once an Interest reaches the LEO segment of the path, routing towards the exit satellite is trivial due to its grid topology, and therefore the complexity is independent of the network size.

Security issues must be also considered, as all traffic travels through intermediate nodes that can inspect it. Fortunately, NDN design already considers these concerns: every Data packet is cryptographically signed by the producer, ensuring integrity; additionally, their content can also be ciphered at the producer, thus providing confidentiality. Moreover, even the suffix parts of the name of the requested data can be ciphered so that only the producer application can decode it.

## 8. Conclusions

Maintaining uninterrupted communication between ground terminals across a massive LEO constellation is challenging for conventional Internet technology due to frequent handovers. This article explores the feasibility of utilizing the NDN architecture to address this issue without requiring modifications to the user software or the NDN protocol itself.

We presented an efficient, scalable method for identifying satellites serving specific geographic locations in order to drive traffic back to ground nodes close to the producers. The method includes the means to reduce the number of satellites receiving the traffic for a given ground node to the actual set being tracked by the ground node.

Additionally, we demonstrated the adaptability of gateways to respond to remote producer mobility while minimizing traffic losses and also successfully adapted an existing consumer mobility protocol for our scenario to address consumer mobility. When taken together, the method for identifying satellites and the producer and consumer mobility solutions provide a complete solution for node mobility in NDN massive LEO constellations.

The results show that our producer mobility solution significantly reduces traffic losses by forwarding Interest packets to two satellites during handover periods, which can be as short as half a second.

Future lines of research need to address the routing of Interest packets in satellite shells with a grid-like topology in more realistic scenarios; for instance, considering link failures and/or incomplete networks. We believe that a low-complexity routing solution can be provided due to the regularity of the scenario. A second future research direction is designing a dynamic method for tuning the *H* parameter that minimizes the handover lengths while keeping the packet losses controlled.

## Figures and Tables

**Figure 1 sensors-26-00309-f001:**
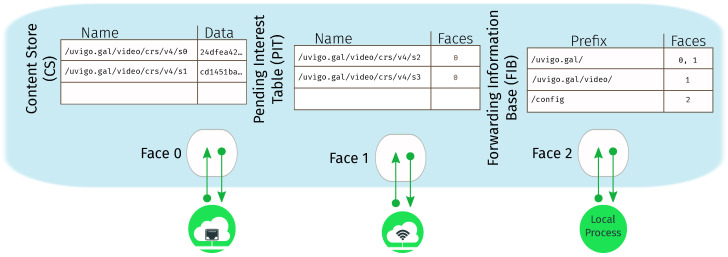
Main forwarding elements in an NDN network layer.

**Figure 2 sensors-26-00309-f002:**
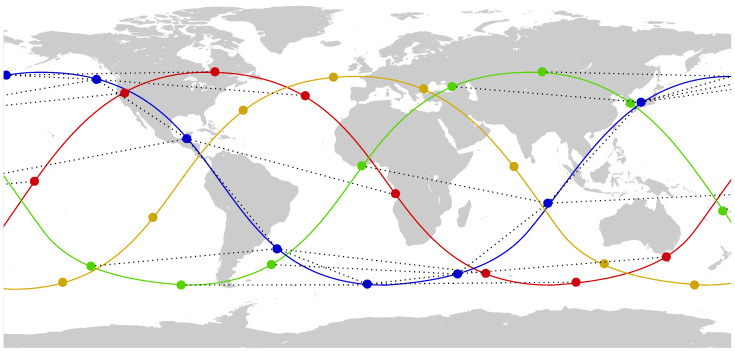
A small constellation with just four orbital planes and eight satellites per plane. The dotted lines show ISL links of *blue* satellites.

**Figure 3 sensors-26-00309-f003:**
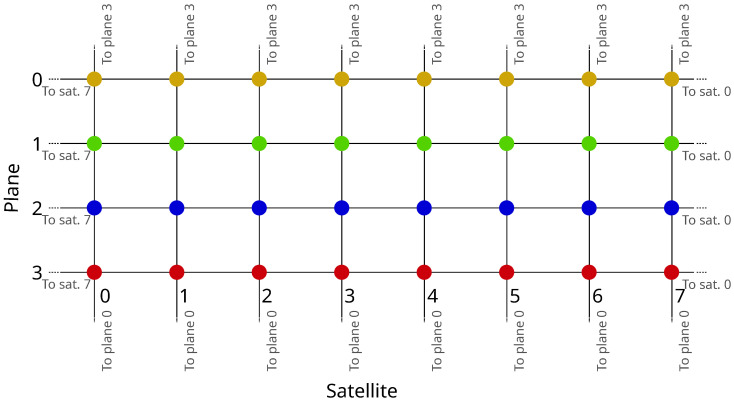
Logical topology of the constellation represented in Figure 2 showing the stable links between satellites.

**Figure 4 sensors-26-00309-f004:**
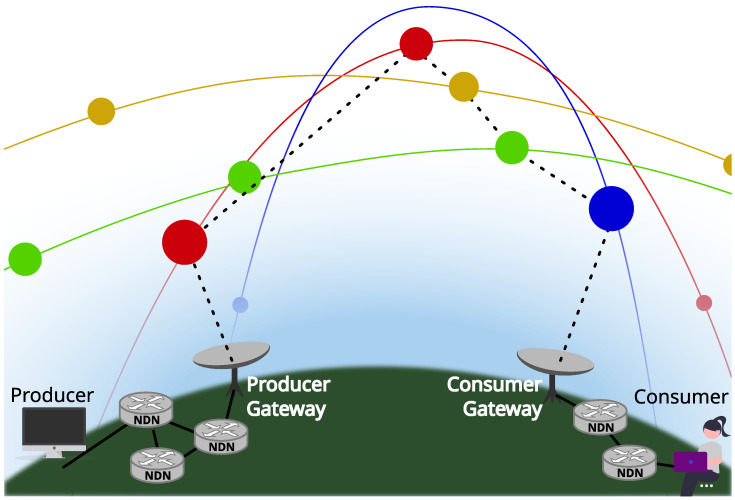
General overview of the scenario.

**Figure 5 sensors-26-00309-f005:**
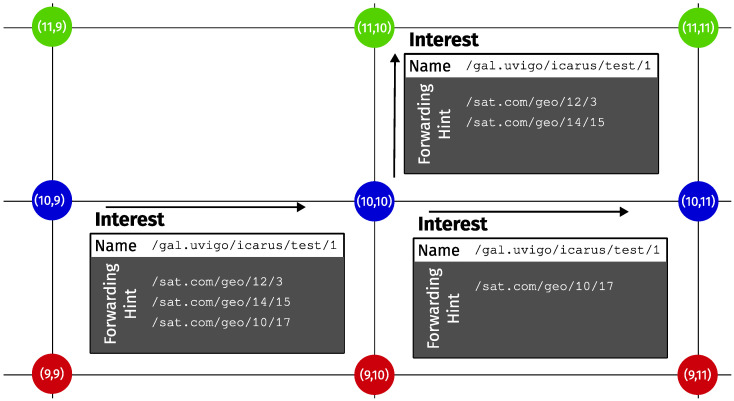
Interest forwarding example.

**Figure 6 sensors-26-00309-f006:**
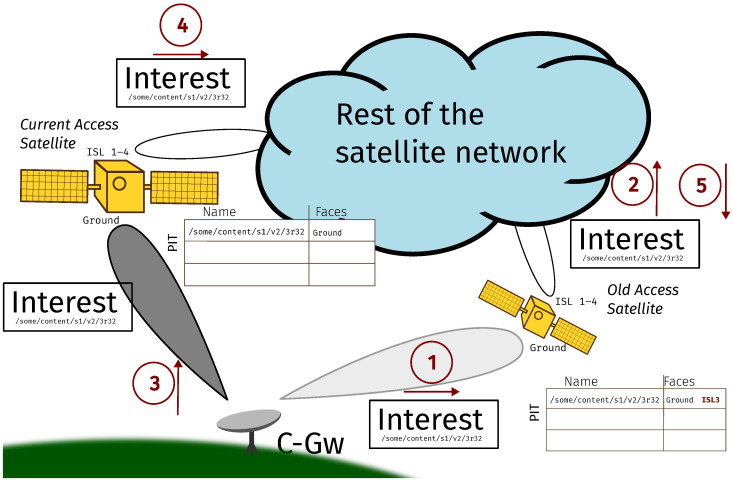
Consumer mobility procedure.

**Figure 7 sensors-26-00309-f007:**
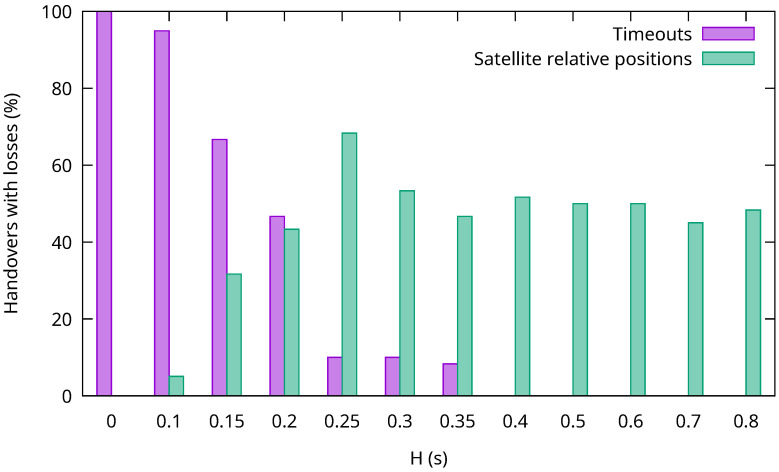
Percentage of P-Gw handovers that suffer packet losses (and their cause) for different handover lengths.

**Figure 8 sensors-26-00309-f008:**
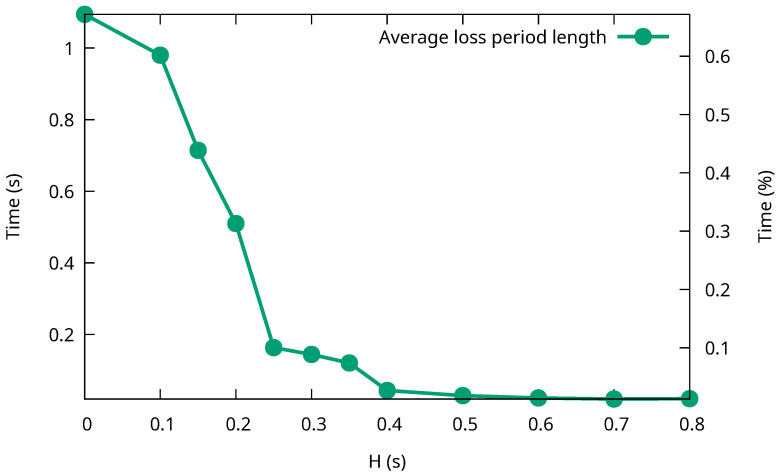
Average length of loss periods after P-Gw satellite handovers.

**Figure 9 sensors-26-00309-f009:**
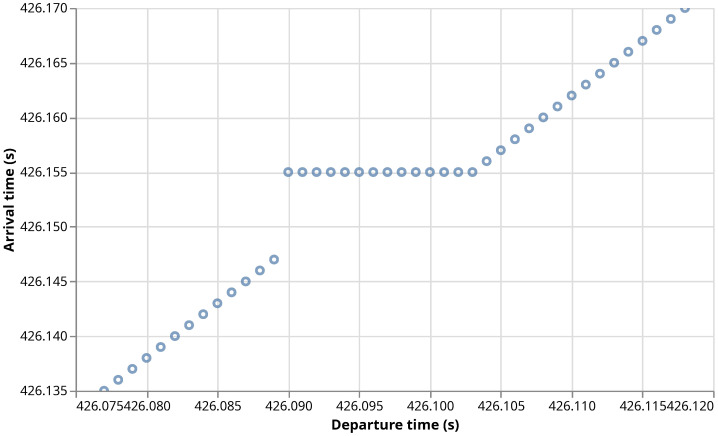
Departure time of Interests and arrival time of the corresponding Data to the C-Gw during a consumer mobility event. Each circle represents the arrival of a Data packet.

## Data Availability

The raw data supporting the conclusions of this article will be made available by the authors on request.

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
