# Peer review of "A Scalable Solution for Node Mobility Problems in NDN-Based Massive LEO Constellations"

_sensors, 2026, doi:10.3390/s26010309_

Round 1

Reviewer 1 Report

Comments and Suggestions for Authors

In this article, the authors present a novel and scalable solution for managing node mobility in Named Data Networking (NDN)-based massive Low Earth Orbit (LEO) satellite constellations, with the goal of enhancing global Internet connectivity. The paper addresses the challenges introduced by frequent handovers and high mobility among ground gateways and satellites in large-scale LEO networks, an environment in which conventional IP mobility protocols often struggle. The proposed solution is based on NDN and leverages its information-centric architecture to efficiently support both consumer and producer mobility without requiring major modifications to the NDN protocol or network.
The experiments conducted by the authors demonstrate a significant reduction in packet loss during handovers, confirming that the proposed approach achieves robust mobility tracking and recovery in large-scale constellations.
Overall, the writing and organization of the paper are satisfactory, and the methodology appears sound. 

However, the following issues should be considered:

1) The proposed solution appears tailored to the specific topology and operational assumptions used in the paper. Its applicability to other satellite architectures or hybrid terrestrial–space networks may require adaptation or additional mechanisms. A discussion of this point should be included.
2) The simulation assumes that ground stations track only one satellite at a time. While this tests a worst-case scenario, it may not reflect multi-beam or multi-link capabilities available in modern ground terminals. The authors should clarify whether this assumption represents a limitation and provide additional justification or discussion.
3) Acronyms should be defined consistently using a uniform format.
4) The conclusion section could be strengthened by including potential future research directions. In addition, citations should not appear in the conclusion.
5) Long dashes ("—") should be replaced with commas, where appropriate, to improve readability and align with formatting conventions.

Reviewer 2 Report

Comments and Suggestions for Authors

The paper “A Scalable Solution for Node Mobility Problems in NDN-based Massive LEO Constellations” approaches the problem of how effectively to manage node mobility in Named Data Networking for massive Low Earth Orbit constellations. The authors propose a scalable solution to enhance connectivity between ground stations during frequent satellite handovers.

The topic is relevant and has original contributions, by addressing a gap in the field of satellite communications, the challenges of mobility in NDN architectures for LEO constellations. The increasing deployment of commercial LEO satellites makes this topic to be very actual.

The technique proposed adds mobility that does not require significant modifications to existing NDN protocols, by introducing a scalable method for identifying ground gateways and optimizing traffic directed towards them.

The results demonstrate that the proposed solution reduces traffic losses and maintains connectivity during handovers.

The text is well written, I see no big problems in terms of language, and the figures have good quality.

I have the following comments/questions that I believe will make the paper more attractive for the readers

  • The text mentions Walker-delta and Walter-star topology. It makes a lot of difference for communications between satellites in neighboring orbital planes, because they have motions in opposite directions. How does your technique deals with this point? Does it work in both cases? What are the consequences of this difference?
  • Regarding the motion of the satellites, which terms did you consider in the dynamics? This is important, because you need to know the locations of the satellites at every moment and this model has strong influence on this.
  • Do you think you need frequent updates in the orbits of the satellites from orbit determination? Or, can you rely on orbit propagations for longer times?
  • Are all the satellites with the same inclination and altitude? If not, the rate of chance of the longitude of the ascending node will not be the same and the geometry of the constellation will change fast in time. How sensible is you technique to this problem?
  • How does your method handle variations in traffic load and network congestion?
  • Are there specific limitations to your approach, in terms of number of satellites and/or orbits of the satellites?
  • Can you explain better the potential integration of your solution with existing network infrastructure?
  • Do you need to make orbital maneuvers to keep the satellites in the same orbit they had when launched? Or orbit decay is not a problem for your technique?
  • Including a comparative analysis with existing solutions could increase the paper's credibility and highlight the advantages of the proposed method more clearly.

Reviewer 3 Report

Comments and Suggestions for Authors
  1. The abstract fails to specify the performance advantage data of the core solution and needs to supplement key indicators (such as packet loss rate, handover delay).
  2. The introduction insufficiently explains why the existing Kite protocol is not applicable to the LEO scenario and needs to elaborate on its defects.
  3. In the "3. Problem Description" section, the specific impact of satellite orbit parameters on handover frequency is not explained, and it is necessary to supplement the correlation analysis.
  4. Section 4.1 does not clarify the synchronization mechanism of the distributed database and needs to explain the method for ensuring data update timeliness.
  5. The applicability of OLC encoding in high-latitude regions has not been verified, and it is necessary to supplement the adaptation description for special scenarios.
  6. The forwarding algorithm (Algorithm 1) does not consider the link fault recovery logic and needs to add a description of the fault-tolerance mechanism.
  7. The basis for determining the optimal value of H in performance optimization is insufficient, and it is necessary to supplement the parameter calibration process in multiple scenarios.
  8. The experimental section does not specify the specific parameters of the satellite node computing power constraints and needs to supplement the simulation configuration details.

Round 2

Reviewer 1 Report

Comments and Suggestions for Authors

The additions to the paper are satisfactory in terms of content. However, the writing and the English of them can be improved.
Please also check for typos, e.g., ‘i.e.’ is not correct.

Reviewer 2 Report

Comments and Suggestions for Authors

I thanks the authors for considering my suggestions and I recommend the paper for publication in its current form.